# Long-Term Exposure to Ozone and Fine Particulate Matter and Risk of Premature Coronary Artery Disease: Results from Genetics of Atherosclerotic Disease Mexican Study

**DOI:** 10.3390/biology11081122

**Published:** 2022-07-27

**Authors:** Rosalinda Posadas-Sánchez, Gilberto Vargas-Alarcón, Andres Cardenas, José Luis Texcalac-Sangrador, Citlalli Osorio-Yáñez, Marco Sanchez-Guerra

**Affiliations:** 1Instituto Nacional de Cardiología Ignacio Chávez, Mexico City 14080, Mexico; rosalinda.posadas@cardiologia.org.mx (R.P.-S.); gilberto.vargas@cardiologia.org.mx (G.V.-A.); 2Division of Environmental Health Sciences, School of Public Health, University of California, Berkeley, CA 94720, USA; andres.cardenas@berkeley.edu; 3Instituto Nacional de Salud Publica, Cuernavaca 62100, Mexico; jtexcalac@insp.mx; 4Departamento de Medicina Genómica y Toxicología Ambiental, Instituto de Investigaciones Biomédicas, Universidad Nacional Autónoma de México, Ciudad Universitaria, Mexico City 04510, Mexico; 5Laboratorio de Fisiología Cardiovascular y Trasplante Renal, Unidad de Investigación en Medicina Traslacional, Instituto de Investigaciones Biomédicas, Universidad Nacional Autónoma de México and Instituto Nacional de Cardiología Ignacio Chávez, Mexico City 14080, Mexico; 6Instituto Nacional de Perinatología, Mexico City 11000, Mexico

**Keywords:** cardiovascular disease, premature coronary artery disease, PM_2.5_, ozone

## Abstract

**Simple Summary:**

Epidemiological studies have identified associations between fine particulate matter (with an aerodynamic diameter of less than 2.5 μm (PM_2.5_)) and ozone exposure with cardiovascular disease; however, studies linking ambient air pollution and premature coronary artery disease (pCAD) in Latin America are nonexistent. We leveraged data from the Genetics of Atherosclerotic Disease (GEA) Mexican study to address the question of the extent to which long-term exposure to ozone and PM_2.5_ exposure was associated with the risk of pCAD. We showed for the first time a higher risk of pCAD associated with 1 ppb increase in ozone (1-year, 2-year, 3-year, and 5-year) and 5μg/m^3^ of PM_2.5_ (5-year) compared to controls. This study provides evidence that ozone and PM_2.5_ may be a modifiable risk factor for pCAD.

**Abstract:**

(1) Background: Epidemiological studies have identified associations between fine particulate matter (PM_2.5_) and ozone exposure with cardiovascular disease; however, studies linking ambient air pollution and premature coronary artery disease (pCAD) in Latin America are non-existing. (2) Methods: Our study was a case–control analysis nested in the Genetics of Atherosclerotic Disease (GEA) Mexican study. We included 1615 participants (869 controls and 746 patients with pCAD), recruited at the Instituto Nacional de Cardiología Ignacio Chávez from June 2008 to January 2013. We defined pCAD as history of myocardial infarction, angioplasty, revascularization surgery or coronary stenosis > 50% diagnosed before age 55 in men and age 65 in women. Controls were healthy individuals without personal or family history of pCAD and with coronary artery calcification equal to zero. Hourly measurements of ozone and PM_2.5_ from the Atmospheric Monitoring System in Mexico City (SIMAT in Spanish; Sistema de Monitero Atmosférico de la Ciudad de México) were used to calculate annual exposure to ozone and PM_2.5_ in the study participants. (3) Results: Each ppb increase in ozone at 1-year, 2-year, 3-year and 5-year averages was significantly associated with increased odds (OR = 1.10; 95% CI: 1.03–1.18; OR = 1.17; 95% CI: 1.05–1.30; OR = 1.18; 95% CI: 1.05–1.33, and OR = 1.13; 95% CI: 1.04–1.23, respectively) of pCAD. We observed higher risk of pCAD for each 5 µg/m^3^ increase only for the 5-year average of PM_2.5_ exposure (OR = 2.75; 95% CI: 1.47–5.16), compared to controls. (4) Conclusions: Ozone exposure at different time points and PM_2.5_ exposure at 5 years were associated with increased odds of pCAD. Our results highlight the importance of reducing long-term exposure to ambient air pollution levels to reduce the burden of cardiovascular disease in Mexico City and other metropolitan areas.

## 1. Introduction

Cardiovascular disease (CVD) is the main cause of mortality globally, accounting for 18.6 million deaths in 2019. According to the Global Burden of Disease study, the CVD burden has continued its decades-long rise in almost all countries, including those in Latin America and the Caribbean (LAC), where prevalent CVD cases are likely to increase because of population growth and aging, among other factors [1]. CAD and its complications are the leading cause of death in men and second in Mexican women [2]. Unfortunately, this deadly disease will remain a prevalent global health threat for the next decades [3,4,5]. Risk factors for CAD include genetic and lifestyle factors and age [6,7]. Typically, CAD risk increases with age because of the higher risk of plaque formation that might lead to CAD clinical manifestations (including angina and myocardial infarction) due to blood flow reduction to the myocardium [7].

CAD is considered premature (pCAD) when a cardiovascular event occurs before 55 years in males and 65 years in females [8,9]. To the best of our knowledge, no previous studies have linked ambient air pollution with pCAD. Most of the studies have focused on the association between air pollution and CAD or cardiovascular mortality [10,11,12,13]. For example, in a case–control study of U.S. residents of Worcester, Massachusetts, an interquartile range (IQR) increase (0.59 µg/m^3^) in PM_2.5_ was associated with a 16% increase in the odds of acute myocardial infarction (AMI) (95% CI: 1.06, 1.29) [14]. Data from the nationwide Danish Nurse Cohort Study on 22,882 female nurses (>44 years), indicated that an IQR increase in PM_2.5_ (3-year running mean) was associated with a higher risk of incident fatal AMI (HR: 1.69; 95% CI: 1.33, 2.13) [15]. In a time-series study conducted in Changzhou, China, PM_2.5_ was associated with an increase of 1.64% (95% CI: 0.54, 2.74%) in the risk of AMI [16]. Overall, studies linking PM_2.5_ and clinical outcomes related to CAD have been conducted mainly in Caucasian or Asian populations, where PM_2.5_ levels, composition and sources are different from those reported in Latin American populations with different demographics and genetic backgrounds [17]. Additionally, the high prevalence of other chronic diseases such as obesity and diabetes in Latin America might potentiate cardiovascular events related to ambient air pollution [18,19].

Ozone is one of the most harmful air pollutants, a powerful oxidizing agent, currently part of air quality guidelines in the U.S., Europe, and Mexico [20]. Since ozone is formed by complex chemical reactions triggered by heat and sunlight, ozone will remain an environmental health concern because of the increases in temperature related to climate change [20,21]. Importantly, the available literature, though sparse, does suggest associations of ozone exposure with cardiovascular outcomes. For example, ozone exposure was associated with an increased rate of carotid wall thickness progression in young adults from six U.S. city regions over almost a decade of follow-up [22]. In a panel study, ozone was associated with alterations across several pathways related to cardiovascular disease such as changes in interleukin-6, monocytes, and large-elasticity index, among others [23].

Mexico City is one of the main megacities in Latin America, and ambient air pollutants such as ozone and PM_2.5_ concentrations exceed national and international guidelines [24,25,26]. However, no previous study has examined associations between ozone and PM_2.5_ with pCAD. Therefore, we aimed to evaluate associations between long-term exposure to ozone and PM_2.5_ and pCAD outcomes in adults from the Genetics of Atherosclerotic Disease (GEA) cohort in Mexico City.

## 2. Materials and Methods

### 2.1. Study Population

We conducted a case–control study nested in the Genetics of Atherosclerotic Disease (GEA) study. The GEA study is a prospective cohort that comprised adults with pCAD and healthy controls without a personal or family history of pCAD [27]. The main goal of this cohort was to elucidate the genetic factors associated with pCAD and other coronary risk factors in the Mexican population. At baseline (2008–2013), subjects were recruited from donors at the blood bank of the National Institute of Cardiology in Mexico City or by advertisements in social service centers. The whole baseline GEA cohort included 2840 individuals, 1240 pCAD patients, and 1600 healthy controls aged from 30 to 75 years [28]. This analysis included 746 patients with premature CAD and 869 controls with available information on air pollution exposure (Figure 1). The study was approved by the institutional review board of the Instituto Nacional de Cardiología Ignacio Chavez (INCICH) (Project number 19-1104) and by the National Institute of Perinatology (project number 2020-1-41). All subjects provided informed consent. We defined pCAD as a history of myocardial infarction, angioplasty, revascularization surgery, or coronary stenosis > 50% (determined by angiography) diagnosed before age 55 in men and before age 65 in women [9]. Participants without acute cardiovascular events in the three months before the study were included and those with congestive heart failure, thyroid and liver disease, kidney cancer, or corticosteroid treatment were not included. Controls were healthy asymptomatic individuals without a personal or family history of pCAD, recruited from the Institute’s blood bank and by direct invitation. In both pCAD patients and controls, chest and abdomen computed tomography was performed and interpreted by trained and experienced radiologists. Scans were read to assess and quantify coronary artery calcification (CAC) score using the Agatston method [29]. Exclusion criteria included renal, thyroid, liver, or oncological disease and congestive heart failure.

All GEA participants answered structured questionnaires that investigated information regarding demographics, family history, medications, smoking, physical activity, and alcohol intake. We measured height and weight to estimate body mass index (BMI). Overweight was defined as BMI ≥ 25 to 30 kg/m^2^, and obesity as BMI > 30 kg/m^2^. Systolic and diastolic blood pressures were measured via a digital sphygmomanometer, Welch Allyn, series 5200 (Skaneateles Falls, NY, USA.), three times after the patient was seated for at least 10 min. The average of the second and third measurements was used for the analysis. Type 2 diabetes mellitus was defined when fasting plasma glucose values were ≥126 mg/dL [30] and was also considered when the patient reported current hypoglycemic drug use or a medical history of type 2 diabetes mellitus.

After at least 10 h of fasting, blood samples from the participants were collected at enrollment time. The glucose, total cholesterol, and high-density lipoprotein cholesterol (HDL-C) concentrations were evaluated in fresh samples, using standardized enzymatic procedures in a Hitachi autoanalyzer 902 (Hitachi LTD, Tokyo, Japan). The accuracy and precision of lipid measurements are constantly evaluated by the Center for Disease Control and Prevention (Atlanta, GA, USA). LDL-C was calculated [31]. We defined smoking status as follows: (a) current smoking when subjects self-reported smoking any tobacco in the previous 12 months, (b) former smokers as those who had quit more than a year earlier. Total physical activity was measured through a standardized and validated questionnaire [32]. We calculated an index of physical activity considering physical activity at work, sports and leisure time as previously described by Beake et al., 1982 [33].

Ambient ozone and PM_2.5_ measures were estimated using the monitoring stations of the National System of Air Quality in 1954 individuals from the entire cohort (1208 healthy individuals and 746 patients with pCAD). CAC score was defined in the 1208 healthy individuals; of these, 869 individuals presented a CAC score equal to zero, and 339 individuals presented a CAC score > zero and were, therefore, excluded as controls and considered as individuals with subclinical atherosclerosis. In the present study, we included 1615 individuals from Mexico City and the metropolitan area, 746 patients with pCAD, and 869 controls with CAC equal to zero (Figure 1).

### 2.2. Air Pollution and Weather Data

Hourly measurements of ambient PM_2.5_ and ozone from 3 October 2003 to 16 December 2012 were obtained from automatic monitoring stations of the National System of Air Quality Information (SINAICA, its acronym in Spanish). We calculated daily averages for PM_2.5_ and 8 h daily maximum concentrations for ozone. We included only daily measurements that met a minimum of 75% completeness of hourly data (18 h by day). We applied spatial analysis and interpolation processes to estimate participants’ exposure using geographic information layers in shapefile format as previously described [34,35]. Residential annual exposure to air pollution (PM_2.5_ or ozone) was assigned to the home address of each study participant following the next steps: (1) For each monitoring station, we constructed 5 and 10 km circular buffers. Then, when houses were located within an intersection of two or more 5 km circular buffers, their air pollution exposures were estimated by squared IDW (inverse distance weighted) interpolation using PM_2.5_ or ozone records from the monitoring stations involved in each intersection; (2) For households not located in 5 km intersection areas, we employed intersection areas for 10 km buffers, and the same method (squared IDW) was used to estimated air pollution exposure; (3) For those households not located in any of the previous intersection areas (5 or 10 km), but within 10 km buffers, we directly assigned air pollution concentration recorded at the monitoring station of such buffer; (4) For the remaining households outside any intersection area or circle buffer; air pollution was estimated using IDW raised to the power value of 1, using all available monitoring stations throughout the city. This four-step method was replicated day by day, and from these results, we calculated annual moving averages.

We obtained hourly climatic data from the SIMAT (Sistema de Monitero Atmosférico de la Ciudad de México) webpage. We calculated daily means of temperature, wind speed, and relative humidity using IDW interpolation for each household location. Then, results were aggregated to obtain annual means. All data processing was performed in R software version 3.6 and RStudio version 0.98.

### 2.3. Statistical Methods

We summarized general characteristics of the study participants using frequencies and percentages or mean and standard deviation. Ozone and PM_2.5_ concentrations were expressed as median and total range. We compared participants’ characteristics for the pCAD group and controls using Chi-Squared tests or Mann–Whitney U.

We selected potential confounders *a priori* hypothesized to influence both air pollution and pCAD. Namely, we adjusted analyses for outdoor temperature (continuous), relative humidity (continuous), wind speed (continuous), sex (female/male), age (continuous), BMI (continuous), education (≤elementary/junior high school/>senior high school), locality (categorical), smoking status(never/former/current), diabetes mellitus (yes/no), HDL-C (continuous), LDL-C (continuous), systolic blood pressure (continuous), antihypertensive medication (yes/no) and physical activity (metabolic equivalent of task-hours per week, continuous).

We fitted multiple logistic regression models to estimate associations between pCAD and long-term exposure to ozone or PM_2.5_ levels across multiple time windows (moving averages from 1 to 5 years). For all of the analyses, we used one ppb increase in ozone and a 5 μg/m^3^ increase in PM_2.5_ levels. In sensitivity analysis, we adjusted for the ambient PM_2.5_ or ozone levels in the matching time window to rule out the possibility that the observed effect was in part attributable to confounding by ambient PM_2.5_ or ozone levels. Additionally, we tested effect modification by BMI and diabetes mellitus; however, we did not test effect modification by sex or age (considered the most influential factors in CAD development) because pCAD definition overlaps with these two concepts [7,8,9].

All statistical analyses were performed using SAS Studio 3.6 (SAS Institute, Cary, NC, USA) and R Study version 3.3.0 (The R Foundation for Statistical Computing, Platform, Vienna, Austria).

## 3. Results

### 3.1. Characteristics of the Study Population

Table 1 shows the characteristics of the study participants (n = 1615). Among patients with pCAD, most were male (80.8%), 45 years or older (88.2%), 83% were overweight or obese, and 53.4% had an elementary school education. A total of 63.9% and 15% of the pCAD participants were former and current smokers, respectively, and 35% had been diagnosed with diabetes mellitus. Compared to controls, pCAD participants were slightly older and less educated, with higher BMI and higher prevalence of diabetes mellitus. pCAD participants were more likely to be former smokers and to use hypertensive medication. Systolic and diastolic blood pressure were statistically significantly higher in pCAD patients compared to controls. Both HDL-C and LDL-C were lower in the pCAD group. LDL-C was lower in pCAD due to statin medication.

### 3.2. Ambient Ozone and PM_2.5_ Levels

We estimated long-term exposure to ozone and PM_2.5_ using moving averages from the first to fifth year before the day of the visit or baseline (Table 2). Median ozone concentrations at 5 years were significantly higher in the pCAD group compared to controls. PM_2.5_ levels were significantly higher in the pCAD group at 3, 4, and 5 years compared to controls.

### 3.3. Association between Air Pollution Levels and pCAD

Compared to controls and after adjusting for potential confounders, ozone (1 ppb increase) was significantly associated with higher odds of pCAD, at 1 year (OR = 1.10; 95% CI: 1.03–1.18); 2 years (OR = 1.17; 95% CI: 1.05–1.30), 3 years (OR = 1.18; 95% CI: 1.05–1.33) and 5 years (OR = 1.13; 95% CI: 1.04–1.23) (Figure 2A). Ozone exposure 4 years before the baseline was also associated with higher odds of pCAD; however, it did not reach statistical significance.

Multivariate analyses also showed a significant association between PM_2.5_ exposure (5 μg/m^3^ increase) at 5 years and pCAD (OR = 2.75; 95% CI: 1.47–5.16) (Figure 2B). Although PM_2.5_ exposures at years 2, 3, and 4 were non-significantly associated with pCAD, we observed a trend moving towards increased odds of pCAD. Associations between ozone exposure (1-year, 2-year, and 3-year) and pCAD remained statistically significant even after PM_2.5_ adjustment at their respective time-windows. However, PM_2.5_ at 5 years and pCAD risk were not further significantly associated after ozone adjustment (Appendix A). Sensitivity analyses showed no effect modification by sex or BMI for pCAD and air pollution associations (data not shown).

All models were adjusted for BMI, age, sex, education, smoking status, diabetes mellitus, HDL-cholesterol, LDL-cholesterol, systolic blood pressure, antihypertensive medication, total physical activity, locality, relative humidity, temperature, and wind velocity. The odds ratio represents the risk of an increase of 1 ppb in ozone or an increase of 5 μg/m^3^ in PM_2.5_. Models of ozone were adjusted for PM_2.5_ in the matching time window, and models of PM_2.5_ were adjusted for ozone levels in the matching time window. Note: Since we have 34 localities, we have shown only one OR per locality.

## 4. Discussion

In this study, we assessed associations between long-term PM_2.5_ and ozone exposure and pCAD in GEA participants, a cardiovascular cohort of Mexican individuals. We observed associations between ozone exposure at 1, 2, 3 and 5 years before the baseline and a higher risk of pCAD. These associations remained significant, except for the 5-year, after adjusting for PM_2.5_. Additionally, we observed a higher risk of pCAD associated with PM_2.5_ at 5 years. Overall, we observed no effect modification by diabetes mellitus or BMI categories either for ozone or PM_2.5_.

Previous studies on ambient air pollution and cardiovascular risk have focused on late clinical manifestations of CAD [14,15], but none have studied PM_2.5_ or ozone related to pCAD. Moreover, population-based cohorts such as MESA [36], CATHGEN [37], ARIC [38], WHI [39], and Nurses’ Health Study [40] linking air pollution and CAD have been conducted in U.S. adults with no or few (8.1% and 22% of Hispanics in WHI and MESA Air, respectively) Latinos. Thus, our results add to the existing literature evidence of pCAD risk related to ambient air pollution in Mexican adults that might have distinct genetic, dietary and environmental factors from those reported in previous studies.

The range of exposures in our study was extremely relevant to other settings. The annual median PM_2.5_ concentration ranged between 21.3 and 27 μg/m^3^, which is at least 4 times higher than the World Health Organization (WHO) annual PM_2.5_ limit of 5.0 μg/m^3^ [41]. Annual concentrations in our study population were higher than those in studies looking at PM_2.5_ and CAD-related outcomes in other areas, such as the CATHGEN study in the U.S., where the majority had an annual PM_2.5_ level of 12.4 μg/m^3^ [37]. Similarly, PM_2.5_ exposure concentrations in our study were higher compared to those reported for the Nurses’ Health Study (mean ± SD: 13.9 ± 2.4 μg/m^3^) [40]. Additionally, the median annual PM_2.5_ concentration in our study was like those in communities classified as high and very high PM_2.5_ exposure in the MESA Study (20 μg/m^3^ and 24 μg/m^3^, respectively) [36].

Studies in the Latin America region are scarce, and most of them focused on associations between PM_2.5_ exposure and cardiopulmonary mortality. For example, a study in Lima, Peru conducted between 2010 to 2016 found positive associations between combined circulatory and respiratory deaths and PM_2.5_ exposure—an increase of 1.8% per 10 μg/m^3^ increase in PM_2.5_ concentration, driven largely by those over 65 years of age [42]. An ecological time-series study conducted in Manaus, Brazil found no significant associations between PM_2.5_ exposure and hospital admissions due to cardiovascular and respiratory diseases in Brazilian children (under 5 years) and the elderly (>60 years) [43]. Finally, a spatial analysis of PM_2.5_ concentrations in Bogotá, Colombia suggested an increase in cardiopulmonary mortality associated with short-term and long-term PM_2.5_ exposure [44]. Overall, previous studies in Latin America focused on cardiovascular disease and air pollution exposure have several limitations: (1) do not consider the specific cause of cardiovascular mortality or hospitalization; (2) ecological designs, and for one of them, the inability to assign exposure estimates at a spatial resolution smaller than the district. All of these limitations might explain discrepancies among the findings [42,43,44].

Our results showed a higher risk of pCAD after 5 years of PM_2.5_ exposure. Previous studies demonstrated that cardiovascular outcomes related to PM_2.5_ largely depend on PM composition [45]. The PM_2.5_ composition in Mexico City has been largely described. PM_2.5_ components include metals (lead, zinc, copper, chromium) [46,47]; polycyclic aromatic hydrocarbons (PAHs) [48]; elemental carbon, organic carbon, and sulfate [49]. These components of PM_2.5_ may contribute to pCAD progression through mechanisms that involve endothelial function and calcium signaling [50,51], inflammation [46,48], and oxidative stress [52,53].

Reactive oxygen species (ROS) production by PM_2.5_ [52] may target key processes of atherosclerosis—the underlying pathophysiology of coronary artery disease (CAD) [54]. For instance, ROS can increase LDLox (pro-atherogenic lipid) formation and concentrations of transcription factors such as TNFalpha and NF-kb. ROS can also decrease nitric oxide (NO, an anti-atherosclerotic molecule) and target mitochondrial calcium signaling, leading to apoptosis—cell death is a component of the plaque necrotic core [51,55].

We found associations between ozone exposure and pCAD risk at different time points. Annual ozone concentrations in our study setting (~80 ppb) were higher compared to those in previous studies in the US, China, and Europe. For example, a study conducted in the Kremnica Mountains in Slovakia during the period 2004–2013 reported maximum O_3_ concentrations of 44.0–50 ppb [56]. In the North China Plain, the COVID-19 lockdown in January 2020 revealed a switch to fast ozone production with maximum daily 8 h ozone concentrations of 60 to 70 ppb [57]. A study conducted in six cities in the eastern US reported May–September annual average concentrations over a 3-year (2013–2015) period in the range of 40–49 ppb [58]. Ozone pollution is another prominent air quality problem in Mexico City [59].

Overall, the link between ozone exposure and CAD outcomes has been less studied than PM_2.5_, and results are mixed [60,61,62,63]. Meta-analyses have shown that ambient ozone exposure is associated with a higher risk of stroke [64], but not myocardial infarction or heart failure [65,66]. In Latin America and the Caribbean region, the ESCALA study (Estudio de Salud y Contaminación del Aire en Latinoamérica) observed that ozone was significantly related to all-cause mortality in Mexico City, Monterrey, Sao Pablo and Rio de Janeiro. However, no associations were observed between ozone and chronic obstructive pulmonary disease and stroke in all ages and the age group ≥ 65 years [67]. Bravo et al. (2016) found no association between increased ozone concentrations and cardiovascular mortality in the population of Sao Paulo, Brazil [68]. Similarly, ozone was not associated with hospital admissions due to diseases of the circulatory system in Santiago, Chile [69].

Our results consistently showed an increased risk of pCAD associated with ozone exposure at 1, 2, 3 and 5 years. The mechanisms that can explain the association between ozone and pCAD might include ROS production, platelet activation, arterial stiffness, blood pressure increase, and changes in fibrinolysis biomarkers [70,71,72]. All of these changes provide the biological plausibility to explain the link between ozone and cardiovascular disease.

Our study has several strengths and limitations; therefore, the results should be interpreted based on those. The strengths of our study include the relatively large cohort with detailed cardiovascular assessment at the individual level, rich covariate data, and long-term ambient air pollution assessment in a highly exposed population. The limitations of our study include the lack of information about air pollution exposure at the place of work or during the commute that might influence personal exposure measures [73]. Additionally, we did not collect information on changes of address during the previous 5 years used to calculate ambient air pollution exposure that might possibly impact PM_2.5_ or ozone levels. We observed differences in key confounders for controls vs. pCAD cases. Therefore, we cannot eliminate the possibility of selection bias. In case–control studies, this selection bias superimposes over the confounding, and it can be controlled in the analyses by the methods used to control for confounding [74]. We adjusted all of our analyses for the key confounders such as sex, age, education, smoking status, diabetes mellitus and physical activity to attenuate the impact of selection bias. Previous studies have demonstrated that some genetic polymorphisms are associated with a higher risk of pCAD in the Mexican-Mestizo population [28,75]. However, we did not evaluate the interaction between genetic susceptibility and air pollution exposure on pCAD risk. The major limitation of our study is its observational nature; therefore, residual or unmeasured confounding cannot be completely ruled out [76]. Despite the limitations, ours is the first study to assess cardiovascular risk in relation to pCAD in Mexican adults exposed to PM_2.5_ and ozone.

## 5. Conclusions

Ambient ozone at different time points and PM_2.5_ exposure at 5 years were associated with an increased risk of pCAD. Our results highlight the importance of reducing ambient air pollution levels to reduce the burden of cardiovascular disease in Mexico City and other large cities across Latin America.

## Figures and Tables

**Figure 1 biology-11-01122-f001:**
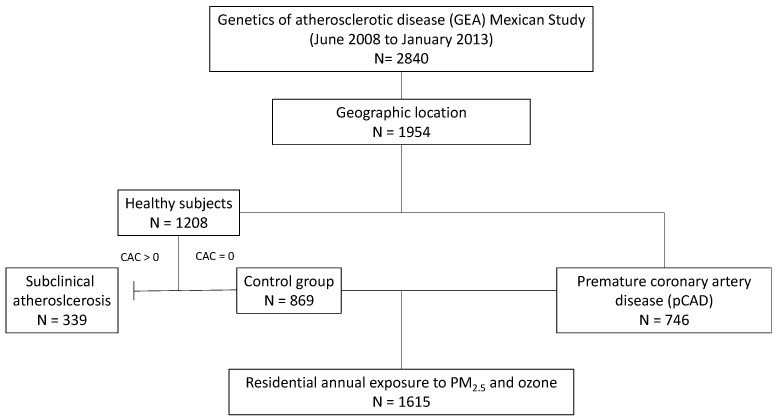
Flowchart for the study participants.

**Figure 2 biology-11-01122-f002:**
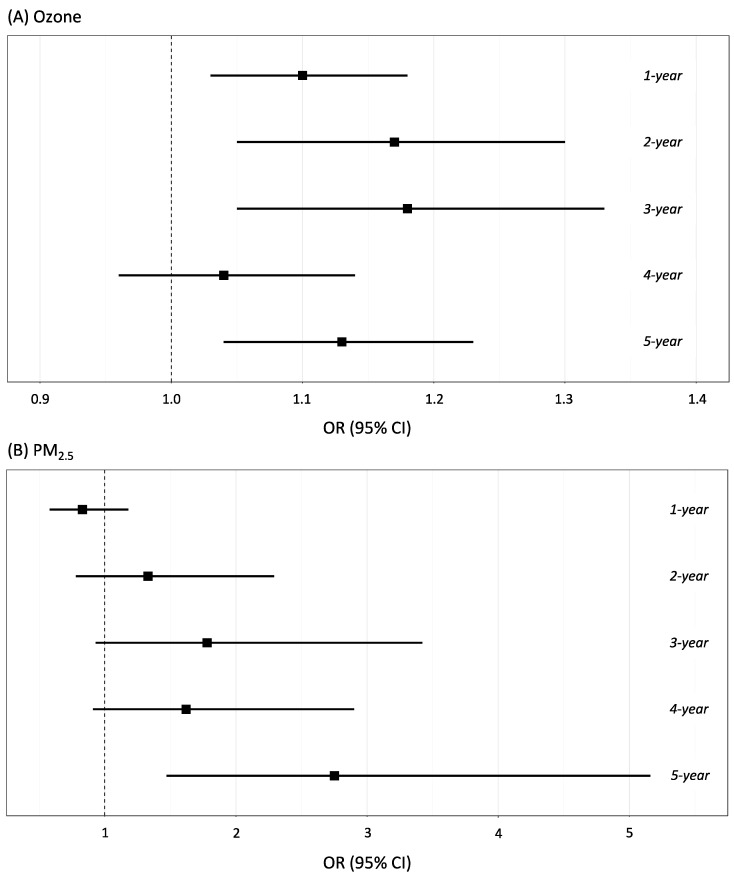
Associations between ozone (**A**) or PM_2.5_ (**B**) levels and pCAD in participants from GEA Study. All models were adjusted for BMI, age, sex, education, smoking status, diabetes mellitus, HDL-cholesterol, LDL-cholesterol, systolic blood pressure, antihypertensive medication, total physical activity, locality, relative humidity, temperature and wind velocity. Odds ratio represents the risk for 1 ppb increase in ozone or 5 μg/m^3^ increase in PM_2.5_.

**Table 1 biology-11-01122-t001:** Characteristics of pCAD patients and controls from the Genetics of Atherosclerotic Disease (GEA) Mexican study.

Characteristics	Control	pCAD	*p*-Value
Overall	869 (53.8%)	746 (46.2%)	
Participant sex			
Male	326 (37.5%)	603 (80.8%)	
Female	543 (62.5%)	143 (19.2%)	<0.0001 *
Age (years)	51.9 ± 9.0	53.9 ± 7.7	<0.0001 ^&^
BMI (kg/m^2^)	28.5 ± 4.5	29.0 ± 4.5	0.005 ^&^
BMI classification (kg/m^2^)			
Normal (18.5–24.9)	193 (22.2%)	127 (17.0%)	
Overweight (25–29.9)	398 (45.8%)	341 (45.7%)	
Obesity (>30.0)	278 (32.0%)	278 (37.3%)	0.01 *
Education			
<Elementary school	257 (29.6%)	398 (53.4%)	
Junior high school	320 (36.8%)	186 (24.9%)	
>Senior high school	292 (33.6%)	162 (21.7%)	<0.0001 *
Cigarette Smoking			
Never smoker	390 (44.9%)	157 (21.1%)	
Former smoker	281 (32.3%)	477 (63.9%)	
Current smoker	198 (22.8%)	112 (15.0%)	<0.0001 *
Diabetes Mellitus			
No	775 (89.2%)	485 (65.0%)	
Yes	94 (10.8%)	261 (35.0%)	<0.0001 *
HDL-C (mg/dL)	46.8 ± 13.8	39.6 ± 10.9	<0.0001 ^&^
LDL-C (mg/dL)	116.0 ± 31.5	98.3 ± 38.0	<0.0001 ^&^
Systolic blood pressure (mmHg)	113.6 ± 15.9	118 ± 18.5	<0.0001 ^&^
Diastolic blood pressure (mmHg)	70.2 ± 8.7	71.79 ± 9.9	0.007 ^&^
Antihypertensive medication			
No	710 (81.7)	247 (33.1)	
Yes	159 (18.3)	499 (66.9)	<0.0001 *
Physical activity	7.8 ± 1.2	7.6 ± 1.3	0.0004 ^&^

Values represent N (%) or mean ± SD. *p*-Values were obtained from Chi-Square test * or Mann–Whitney U Test ^&^.

**Table 2 biology-11-01122-t002:** Ambient ozone levels and PM_2.5_ concentrations of the study participants: pCAD or controls.

	Total (N = 1615)	Controls (N = 869)	pCAD (N = 746)	*p*-Value
Ozone (ppb)				
1-year	75.8 (68.5–81.2)	75.9 (68.7–81.2)	75.7 (68.5–80.8)	0.22
2-year	75.6 (71.4–82.6)	75.5 (71.4–82.6)	75.7 (71.5–81.8)	0.38
3-year	76.5 (71.8–83.1)	76.3 (71.8–83.1)	76.6 (71.8–81.9)	0.05
4-year	77.5 (73.1–84.9)	77.3 (73.1–84.9)	77.6 (73.2–83.6)	0.14
5-year	78.6 (73.7–84.9)	78.4 (73.7–84.9)	78.9 (73.7–84.7)	0.02
PM_2.5_ (μg/m^3^)				
1-year	24.6 (17.7–31.7)	24.7 (17.8–29.9)	24.5 (17.7–31.7)	0.05
2-year	23.9 (20.8–29.6)	23.9 (21.1–29.6)	23.9 (20.8–29.2)	0.76
3-year	23.6 (21.5–29.5)	23.5 (21.6–29.3)	23.7 (21.5–29.5)	<0.0001
4-year	24.2 (21.9–29.8)	24.0 (21.9–29.7)	24.3 (21.9–29.8)	<0.0001
5-year	24.7 (22.3–29.7)	24.5 (22.3–29.7)	25.0 (22.3–29.7)	<0.0001

Values represent median and total range (min-max). *p*-Values were obtained from Mann-Whitney U Test.

## Data Availability

Not applicable.

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
