# Peer review of "Long-Term Exposure to Ozone and Fine Particulate Matter and Risk of Premature Coronary Artery Disease: Results from Genetics of Atherosclerotic Disease Mexican Study"

_biology, 2022, doi:10.3390/biology11081122_

Round 1

Reviewer 1 Report

Please see the attached word document for full comments and review.

Author Response

Posadas-Sánchez et al., Ref: biology-1775168

Long-term exposure to ozone and fine particulate matter and risk of premature coronary artery disease: results from Genetics of Atherosclerotic Disease Mexican study

Section Managing Editor's Remarks to Author

Please use the version of your manuscript found at the above link for your revisions. 

(I) Please check that all references are relevant to the contents of the manuscript.
(II) Any revisions to the manuscript should be marked up using the “Track Changes” function if you are using MS Word/LaTeX, such that any changes can be easily viewed by the editors and reviewers. 
(III) Please provide a cover letter to explain, point by point, the details of the revisions to the manuscript and your responses to the referees’ comments. 
(IV) If you found it impossible to address certain comments in the review reports, please include an explanation in your rebuttal.
(V) The revised version will be sent to the editors and reviewers.

If one of the referees has suggested that your manuscript should undergo extensive English revisions, please address this issue during revision. We propose that you use one of the editing services listed at https://www.mdpi.com/authors/englishor have your manuscript checked by a native English-speaking colleague.

RESPONSE TO REVIEWER’S COMMENTS

We greatly appreciate the thoughtful and constructive recommendations from the editor and the reviewers. We have addressed them thoroughly and believe that our manuscript has been strengthened as a result. Below we enumerate the changes made in response to reviewer’s suggestions.

Comments from the Editors and Reviewers:

Response to Reviewer 1 Comments

Overview

The authors present an interesting study that utilized a rich existing dataset from June 2008 – January 2013 (Genetics of Atherosclerotic Disease Mexican Study) and hourly ambient air pollution data (ozone and PM2.5) calculated for annual estimates, collected throughout Mexico City. Careful selection criteria with scans read by trained radiologists were conducted for both cases and controls. In-depth additional variables included factors such as other health measures, demographics, and smoking.  Key findings suggested increased odds of pCAD among men and women associated with increased long-term exposure to ambient air pollution. Ozone seemed to have more immediate effects, evident with increased odds of pCAD each year, while PM2.5 seemed to show adverse effects only at the 5-year mark, although there was a signal of increased odds at years 3 and 4. There are some minor edits needed throughout the paper, and a few places for greater discussion regarding interpreting results and addressing limitations. However, overall, this paper is well-written and a unique contribution to the literature, especially with its focus on pCAD among a Mexican population.

Response: Thanks for the careful review and we appreciate your constructive comments. We made changes to the manuscript using Reviewer #1’s suggestions in the revised manuscript, and we believe the manuscript has been strengthened as a result.

Minor edits

  1. 1, line 30: use delete “fine particulate matter” and just use abbreviation of PM2.5 since you have already defined it in first line.

Response:  Thank you for this comment. Change made as suggested.

P.1, line 32: confusing wording “each ppb increase…” does this mean to say you show an incremental increase in ozone (ppb) per year? And why does this not mention PM, just ozone?

Response: We made some changes for clarity as follows:

Simple Summary: Page 1, lines 30-32: “We showed for the first time a higher risk of pCAD associated with 1 ppb increase in ozone (1-year, 2-year, 3-year, and 5-year) and 5mg/m3 of PM2.5 (5-year) compared to controls.”

  1. 1, line 39: are these studies in Latin America “scarce,” like you say in Simple Summary, or “non-existing?” Need to be consistent.

Response: We agree with the reviewer, we must be consistent.

Simple Summary: Pages 1, lines 27-28: “Studies linking PM2.5 exposure and premature coronary disease in Latin America are non/existing.”

  1. 3, lines 61-63: Need to update the GBD statistics with their most current reports, 2020 or later, including global statistics and focus on Latin America for CVD.

Response: We updated the references as suggested by the reviewer.

Page 2, lines 57-61: “Cardiovascular disease (CVD) is the main cause of mortality globally accounting for 18.6 million deaths in 2019. According to the Global Burden of Disease study, CVD burden continues its decades-long rise in almost all countries, including Latin America and the Caribbean (LAC), where prevalent CVD cases are likely to increase because of population growth and aging, among other factors [1].”

  1. 3, line 67: “thread” should be “threat.”

Response: Sorry for this oversight. Change made as suggested.

  1. 4, line 137: define which device was used for blood pressure measurements?

Response: We added the name of the device as suggested

Materials and Methods: Page 3, lines 131-134: “Systolic and diastolic blood pressures were measured via a digital sphygmomanometer, Welch Allyn, series 5200 (Skaneateles Falls, New York, USA.), three times after the patient was seated for at least 10 minutes.”

  1. 4, line 138: “sited” should be “seated.”

Response: Sorry for this oversight. Change made as suggested.

  1. 4, line 151: more details needed for physical activity measures since it’s such an important variable.

Response: We added more information on physical activity questionnaire as suggested by the reviewer.

Materials and Methods: Pages 3, lines 146-149: “Total physical activity was measured through a standardized and validated questionnaire [2]. We calculated an index of physical activity considering physical activity at work, sports, and leisure time as previously described by Beake et al. 1982 [3].”

  1. 5, Figure 1: “Geografic” should be “Geographic”

Response: Change made as suggested. Thank you

  1. 5, line 171: were all cases and controls present for ambient pollution interpolation for the entire 5-year study period? In other words, was the 5-year ambient air pollution data relevant for all participants, or did some participants live elsewhere during that window?

Response: The reviewer raises an important point. We assumed that all participants lived at the same address for the entire 5/year study period. However, we cannot discard that some participants who might move to a different address during that period. Therefore, we added as a limitation of the study this point as follows:

Discussion: Page 12, lines 371-376: “The limitations of our study include the lack of information about air pollution exposure in the place of work or commute that might influence personal exposure measures [4]. Additionally, we did not collect information on changes of address during the previous 5 years used to calculate ambient air pollution exposure that might possibly impact PM2.5 or ozone levels.

  1. 5, line 199: can use abbreviation “BMI” for body mass index since you have already defined it previously.

Response: Thank you, change was made as suggested.

  1. 6, line 210: If you tested for effect modification of BMI and diabetes, need to explain here.

Response: We added a sentence explaining this point as suggested by the reviewer.

Statistical methods: Page 5, line 212: “Additionally, we tested effect modification by BMI and diabetes mellitus”

  1. Results: does the journal request bullet point paragraphs for Results? If not, I suggest staying consistent with your formatting throughout the manuscript and avoiding bullet point paragraphs.

Response: We agree with the reviewer, our format must be consistent. We deleted bullet point paragraphs. Thank you

  1. 8, Figure 2A and 2B: Very nice figures but the text is too small to read, please increase font size and clarity.

Response: Thank you, we increased font size and clarity as suggested by the reviewer.

  1. 10, lines 317 and 325: subscript PM2.5.

Response: Changes made as suggested.

Broader items to address

  1. Were all cases and controls living in the study area (Mexico City) for the entire 5-year exposure period?

Response 1: We greatly appreciate your concern and are very thankful for your extremely constructive question. In the revised manuscript, we discussed this briefly as a limitation of the study.

Discussion: Page 12, lines 371-376: “The limitations of our study include the lack of information about air pollution exposure in the place of work or commute that might influence personal exposure measures [4]. Additionally, we did not collect information on changes of address during the previous 5 years used to calculate ambient air pollution exposure that might possibly impact PM2.5 or ozone levels”.

  1. There is some concern with the selection of the controls as being non-representative of the sources of the cases. Finding a suitable control group is always a crux of any case-control design. In this sample, the controls do not seem to be a representative sample of the cases because they significantly differ in all of the key confounders- controls were more likely to be female, slightly younger, higher education, nonsmokers, no diabetes, better blood lipid panels, slightly more physical activity compared to cases. This is not something the authors could change since it was a study design issue but needs to be more thoroughly addressed as an important weakness and limitation in the study and dig into how this imbalance might impact results.

Response 2: This is a very important point. We might have some selection bias and we agree with the reviewer that we need to discuss this point.

Discussion: Pages xx, lines x-x: “We observed differences in key confounders for controls vs. pCAD cases. Therefore, we cannot eliminate the possibility of selection bias. In case-control studies, this selection bias superimposes over the confounding and it can be controlled in the analyses by the methods used to control for confounding [5]. We adjusted all our analyses for these key confounders such as sex, age, education, smoking status, diabetes mellitus and physical activity to attenuate the impact of selection bias.”

One positive note is that there are no real differences in exposure measures between controls vs. cases for the 5-year window for either air pollutant. The significant p-value is an artifact of such large sample sizes, but actual differences in ozone and PM between the two groups are not meaningful.

Response: This is true, we observed no such big differences in PM2.5 or ozone levels between the two groups. For some time/points, we observed a slight increase in PM2.5 or ozone for pCAD cases compared to controls. We hypothesized that ambient air pollution might not be the only factor needed to increase CVD risk, however, populations with cardiovascular risk due to lifestyle factors such as overweight obesity, aging, and high lipid levels can be more susceptible to the cardiovascular effects related to ambient air pollution. Thus, even small increases in these contaminants might have an impact on the total cardiovascular risk. However, we need more studies to test this hypothesis and probably a different study design like cohort studies following healthy people vs. people with some cardiovascular risk factors and test the incidence of CVD related to ambient air pollution over time in these two groups. Thank you for the thoughtful review of our manuscript.

  1. Even though PM results at year 3 and 4 were “non-significant” with the 95% CI including the null value, there is a signal of a trend moving towards increased odds of pCAD given higher levels of PM. Worth noting.

Response 3: In this version of the paper, we mentioned the trend towards increased odds of pCAD for pM2.5 at years 2, 3, and 4 as suggested by the reviewer. Thanks

Results: Page 7, lines 256-258: Although PM2.5 exposures at years 2 3, and 4 were non-significantly associated with pCAD, we observed a trend moving towards increased odds of pCAD.

Reviewer 2 Report

In this study by Dr. Marco Sanchez-Guerra and colleagues, the authors concluded that Ozone exposure at different time-points and PM2.5 exposure at 5 years were associated with increased odds of pCAD. This is an interesting study addition to previous art about the relationship of exposure to ambient air pollution levels and the risk of premature coronary artery disease especially in Mexico City; however, I have a few concerns about the following issues which may help to strengthen the conclusions, to improve the scientific rigor and significance.

1. It’s really nice to see the association between air pollution levels and pCAD. The Table(s) should be improved to show the whole logistic regression analysis, especially the P values, OR, and 95%CI after adjusted including showing different risk factors and parameters, which could be much easier to see which parameter(s) is/are the key contributor(s). 

2. The table about the characteristics of study participants should include more details, especially the medication history (such as hypertension related medication, or please claim whether they took medication while testing BP-value) and other Infos. And other related parameters should be included in the adjusted logistic regression.

3. Please clarify what statistical methods were used for composition ratios in clinical data. 

Minor Comments:

1. Proof-reading for the language by a native speaker is needed, including the grammatical errors and some spelling errors. Also, acronyms should be defined the first time they are mentioned.

2. Please double-check all the units in the Table and make sure it’s not missing.

Author Response

Posadas-Sánchez et al., Ref: biology-1775168

Long-term exposure to ozone and fine particulate matter and risk of premature coronary artery disease: results from Genetics of Atherosclerotic Disease Mexican study

Section Managing Editor's Remarks to Author

Please use the version of your manuscript found at the above link for your revisions. 

(I) Please check that all references are relevant to the contents of the manuscript.
(II) Any revisions to the manuscript should be marked up using the “Track Changes” function if you are using MS Word/LaTeX, such that any changes can be easily viewed by the editors and reviewers. 
(III) Please provide a cover letter to explain, point by point, the details of the revisions to the manuscript and your responses to the referees’ comments. 
(IV) If you found it impossible to address certain comments in the review reports, please include an explanation in your rebuttal.
(V) The revised version will be sent to the editors and reviewers.

If one of the referees has suggested that your manuscript should undergo extensive English revisions, please address this issue during revision. We propose that you use one of the editing services listed at https://www.mdpi.com/authors/englishor have your manuscript checked by a native English-speaking colleague.

RESPONSE TO REVIEWER’S COMMENTS

We greatly appreciate the thoughtful and constructive recommendations from the editor and the reviewers. We have addressed them thoroughly and believe that our manuscript has been strengthened as a result. Below we enumerate the changes made in response to reviewer’s suggestions.

Response to Reviewer 2 Comments

In this study by Dr. Marco Sanchez-Guerra and colleagues, the authors concluded that Ozone exposure at different time-points and PM2.5 exposure at 5 years were associated with increased odds of pCAD. This is an interesting study addition to previous art about the relationship of exposure to ambient air pollution levels and the risk of premature coronary artery disease especially in Mexico City; however, I have a few concerns about the following issues which may help to strengthen the conclusions, to improve the scientific rigor and significance. 

  1. It’s really nice to see the association between air pollution levels and pCAD. The Table(s) should be improved to show the whole logistic regression analysis, especially the P values, OR, and 95%CI after adjusted including showing different risk factors and parameters, which could be much easier to see which parameter(s) is/are the key contributor(s). 

Response 1: We thank the reviewer for pointing this out and appreciate your constructive comments. We modified Supplemental Table 1 as suggested to include other factors.

1-year

2-year

3-year

4-year

5-year

OR (95% CI)

p-Value

OR (95% CI)

p-Value

OR (95% CI)

p-Value

OR (95% CI)

p-Value

OR (95% CI)

p-Value

Ozone

1.20 (1.11 – 1.31)

<0.0001

1.19 (1.05 – 1.35)

0.006

1.16 (1.02 – 1.33)

0.02

0.99 (0.88 – 1.11)

0.89

1.07 (0.97 – 1.19)

0.18

PM2.5

0.46 (0.29 – 0.72)

0.0007

0.83 (0.44 – 1.57)

0.56

1.14 (0.54 – 2.43)

0.73

1.68 (0.78 – 3.61)

0.18

1.97 (0.89 – 4.36)

0.09

Relative humidity

0.98 (0.94 – 1.03)

0.45

1.09 (1.01 – 1.19)

0.02

0.96 (0.88 – 1.05)

0.39

1.05 (0.95 – 1.17)

0.30

1.01 (0.90 – 1.13)

0.85

Temperature

1.49 (1.12 – 1.98)

0.006

1.77 (1.18 – 2.67)

0.006

1.20 (0.73 – 2.00)

0.47

0.68 (0.40 – 1.17)

0.17

0.91 (0.53 – 1.55)

0.72

Wind velocity

0.62 (0.28 – 1.34)

0.22

0.60 (0.19 – 1.83)

0.37

0.44 (0.13 – 1.50)

0.19

0.27 (0.08 – 0.92)

0.04

0.57 (0.16 – 1.99)

0.38

BMI

0.95 (0.92 – 0.99)

0.007

0.96 (0.92 – 0.99)

0.01

0.96 (0.93 – 0.99)

0.02

0.96 (0.93 – 0.99)

0.02

0.96 (0.93 – 0.99)

0.02

Sex

10.30 (7.12 – 15.11)

<0.0001

10.47 (7.27 – 15.30)

<0.0001

10.08 (7.01 – 14.71)

<0.0001

10.08 (7.01 – 14.70)

<0.0001

10.13 (7.00 – 14.66)

<0.0001

Age

1.01 (0.99 – 1.03)

0.34

1.01 (0.99 – 1.03)

0.14

1.01 (0.99 – 1.03)

0.18

1.01 (1.00 – 1.03)

0.11

1.01 (0.99 – 1.03)

0.14

Junior high school

0.45 (0.31 – 0.65)

<0.0001

0.44 (0.30 – 0.64)

<0.0001

0.46 (0.32 – 0.66)

<0.0001

0.46 (0.32 – 0.66)

<0.0001

0.47 (0.32 – 0.68)

<0.0001

> Senior high school

0.26 (0.17 – 0.39)

<0.0001

0.27 (0.18 – 0.39)

<0.0001

0.27 (0.18 – 0.39)

<0.0001

0.27 (0.18 – 0.40)

<0.0001

0.27 (0.18 – 0.40)

<0.0001

Former smoker

2.59 (1.83 – 3.68)

<0.0001

2.58 (1.83 – 3.66)

<0.0001

2.58 (1.83 – 3.65)

<0.0001

2.63 (1.87 – 3.71)

<0.0001

2.59 (1.84 – 3.66)

<0.0001

Current smoker

1.11 (0.72 – 1.71)

0.62

1.12 (0.73 – 1.72)

0.59

1.13 (0.74 – 1.72)

0.58

1.13 (0.74 – 1.72)

0.58

1.11 (0.73 – 1.70)

0.62

Diabetes Mellitus

3.51 (2.38 – 5.21)

<0.0001

3.67 (2.50 – 5.44)

<0.0001

3.72 (2.54 – 5.50)

<0.0001

3.71 (2.53 – 5.50)

<0.0001

3.76 (2.56 – 5.58)

<0.0001

HDL-cholesterol

0.98 (0.97 – 0.99)

0.003

0.98 (0.97 – 0.99)

0.006

0.98 (0.97 – 0.99)

0.01

0.98 (0.97 – 0.99)

0.01

0.98 (0.97 – 0.99)

0.008

LDL-cholesterol

0.99 (0.98 – 0.99)

<0.0001

0.99 (0.98 – 0.99)

<0.0001

0.99 (0.98 – 0.99)

<0.0001

0.99 (0.98 – 0.99)

<0.0001

0.99 (0.98 – 0.99)

<0.0001

Systolic Blood Pressure

0.98 (0.97 – 0.99)

0.05

0.98 (0.97 – 0.99)

0.0001

0.98 (0.97 – 0.99)

<0.0001

0.98 (0.97 – 0.99)

<0.0001

0.98 (0.97 – 0.99)

<0.0001

Antihypertensive medication

12.93 (9.12 – 18.61)

<0.0001

13.01 (9.19 – 18.69)

<0.0001

13.27 (9.38 – 19.04)

<0.0001

13.34 (9.44 – 19.13)

<0.0001

13.65 (9.64 – 19.60)

<0.0001

Physical activity

0.89 (0.79 – 1.00)

<0.0001

0.89 (0.79 – 1.00)

0.05

0.89 (0.80 – 1.00)

0.06

0.90 (0.80 – 1.00)

0.06

0.89 (0.80 – 1.00)

0.06

Locality

0.45 (0.12 – 1.59)

0.22

0.62 (0.16 – 2.37)

0.49

0.70 (0.18 – 2.76)

0.61

0.59 (0.15 – 2.37)

0.46

0.88 (0.21 – 3.63)

0.85

Supplemental Table 1 Sensitivity analysis of the associations between ozone or PM2.5 and pCAD in participants from GEA Study. All models were adjusted for BMI, age, sex, education, smoking status, diabetes mellitus, HDL-cholesterol, LDL-cholesterol, systolic blood pressure, antihypertensive medication, total physical activity, locality, relative humidity, temperature, and wind velocity. Odds ratio represents the risk for an increase in 1 ppb in ozone or for an increase in 5 of PM2.5 mg/m3. Models of ozone were adjusted for PM2.5 in the matching time window and models of PM2.5 were adjusted for ozone levels in the matching time window. Note: Since we have 34 localities, we have shown only one OR for locality.

  1. The table about the characteristics of study participants should include more details, especially the medication history (such as hypertension related medication, or please claim whether they took medication while testing BP-value) and other Infos. And other related parameters should be included in the adjusted logistic regression.

Response 2: We ran more analyses following the suggestions from REVIEWERS #1 and #2 and the results have changed since we included Antihypertensive medication as a covariate. We have included all those results in this version and made the respective changes in the whole manuscript.

Table 1. Characteristics of study participants included from the Genetics of Atherosclerotic Disease (GEA) Mexican study

Characteristics

Control

pCAD

P-value

Overall

869 (53.8%)

746 (46.2%)

Participant sex

   Male

326 (37.5%)

603 (80.8%)

   Female

543 (62.5%)

143 (19.2%)

<0.0001*

Age (years)

51.9 ± 9.0

53.9 ± 7.7

<0.0001&

BMI (kg/m2)

28.5 ± 4.5

29.0 ± 4.5

0.005&

Education

   <Elementary school

257 (29.6%)

398 (53.4%)

   Junior high school

320 (36.8%)

186 (24.9%)

   > Senior high school

292 (33.6%)

162 (21.7%)

<0.0001*

Cigarette Smoking

   Never smoker

390 (44.9%)

157 (21.1%)

   Former smoker

281 (32.3%)

477 (63.9%)

   Current smoker

198 (22.8%)

112 (15.0%)

<0.0001*

Diabetes Mellitus

No

775 (89.2%)

485 (65.0%)

Yes

94 (10.8%)

261 (35.0%)

<0.0001*

HDL-C (mg/dL)

46.8 ± 13.8

39.6 ± 10.9

<0.0001&

LDL-C (mg/dL)

116.0 ± 31.5

98.3 ± 38.0

<0.0001&

Systolic Blood Pressure (mmHg)

113.6 ± 15.9

118 ± 18.5

<0.0001&

Diastolic Blood Pressure (mmHg)

70.2 ± 8.7

71.79 ± 9.9

0.007&

Antihypertensive medication

No

710 (81.7)

247 (33.1)

Yes

159 (18.3)

499 (66.9)

<0.0001*

Physical activity

7.8 ± 1.2

7.6 ± 1.3

0.0004&

Values represents N (%) or mean ± SD

P-Values were obtained from Chi-Square test* or Mann-Whitney U Test&.

  1. Please clarify what statistical methods were used for composition ratios in clinical data. 

Response 3: We have clarified this in Table 1 “P-Values were obtained from Chi-Square test* or Mann-Whitney U Test&.”

Minor Comments:

  1. Proof-reading for the language by a native speaker is needed, including the grammatical errors and some spelling errors. Also, acronyms should be defined the first time they are mentioned.

Response: We thank the reviewer for this observation. We made several changes to the manuscript.

  1. Please double-check all the units in the Table and make sure it’s not missing.

Response:  Thank you for the suggestion.

Round 2

Reviewer 2 Report

Thank you for your responses. The manuscript has been nicely revised with new convincing data added.